# Phenotype-Based Genetic Analysis Reveals Missing Heritability of *KIF11*-Related Retinopathy: Clinical and Genetic Findings

**DOI:** 10.3390/genes14010212

**Published:** 2023-01-13

**Authors:** Haoyu Chang, Xin Zhang, Ke Xu, Nien Li, Yue Xie, Weiyu Yan, Yang Li

**Affiliations:** Beijing Ophthalmology & Visual Sciences Key Lab, Beijing Institute of Ophthalmology, Beijing Tongren Eye Center, Beijing Tongren Hospital, Capital Medical University, Beijing 100051, China

**Keywords:** *KIF11*, deep intronic variant, minigene assay, microcephaly with or without chorioretinopathy, lymphedema, or mental retardation, familial exudative vitreoretinopathy

## Abstract

The purpose of this study was to detect the missing heritability of patients with *KIF11*-related retinopathy and to describe their clinical and genetic characteristics. We enrolled 10 individuals from 7 unrelated families harboring a pathogenic monoallelic variant in *KIF11*. All subjects underwent ophthalmic assessment and extraocular phenotype evaluations, as well as comprehensive molecular genetic analyses using next-generation sequencing. Minigene assays were performed to observe the effects of one novel deep intron variant (DIV) and one novel synonymous variant on pre-mRNA splicing. We detected 6 novel different disease-causing variants of *KIF11* in the seven pedigrees. Co-segregation analysis and ultra-deep sequencing results indicated that 5 variants arose de novo in 5 families (71%). Functional validation revealed that the synonymous variant leads to an exon skip, while the DIV causes a pseudoexon (PE) inclusion. The patients presented with high variations in their phenotype, and two families exhibited incomplete penetrance. Ocular manifestations and characteristic facial features were observed in all patients, as well as microcephaly in seven patients, intellectual disability in five patients, and lymphedema in one patient. The key retinal features for *KIF11*-related retinopathy were retinal folds, tractional retinal detachment, and chorioretinal dysplasia. All seven probands had more severe visual detects than other affected family members. Our findings widen the genetic spectrum of *KIF11* variants. DIV explained rare unresolved cases with *KIF11*-related retinopathy. The patients displayed a variable phenotype expressivity and incomplete penetrance, indicating the importance of genetic analysis for patients with *KIF11*-related retinopathy.

## 1. Introduction

The *KIF11* gene (GenBank NM_004523, OMIM 148760) is located on chromosome 10q23.33 and encodes a motor protein, KIF11, also known as Eg5 or Kinesin-5. KIF11, which belongs to the kinesin family, plays a role in cell division and is involved in physiological and pathological growth of the vascular system function [1,2]. The pathogenic variants in *KIF11* were first identified in patients with microcephaly, chorioretinal dysplasia, and primary lymphedema syndrome (MCLMR, OMIM 152950) [3,4,5]. This rare syndrome is inherited in an autosomal dominant pattern and is characterized by microcephaly, ocular anomalies (including chorioretinopathy), congenital lymphedema of the lower limbs, and mild to moderate mental retardation. Patients with MCLMR usually present special facial appearances, including upslanting palpebral fissures, a broad nose with a rounded tip, a long philtrum and a thin upper lip, a prominent chin, and prominent ears [3,4,5]. Pathogenic variants in *KIF11* were subsequently detected in patients with familial exudative vitreoretinopathy (FEVR, OMIM 133780) [6,7].

FEVR is a rare hereditary developmental disorder associated with abnormal retinal vasculature. Its complicated and variable retinal manifestations can include non-perfusion in the peripheral retina, falciform retinal folds, retrolenticular fibrotic masses, and tractional retinal detachment (RD) [8]. FEVR can be inherited in autosomal dominant, recessive, and X-linked patterns and can occur with several other disease-causing genes (LRP5, FZD4, TSPAN12, NDP, CTNNA1, CTNNB1, and ZNF408) in addition to *KIF11* [8]. Several previous studies found that some patients with FEVR carrying *KIF11* variants also had syndromic features, such as microcephaly and mental retardation, indicating a complex phenotypic overlap between MCLMR and FEVR [6,7,9,10,11]. Therefore, the term *KIF11*-associated retinopathy has been proposed to refer these disorders by genotype rather than phenotype [12,13,14].

At present, 122 pathologic *KIF11* variants have been listed according to the Human Gene Mutation Database (HGMD Professional 2021.4). The majority of *KIF11* variants are truncating variants, such as nonsense and frameshift indels, and they introduce a premature termination codon (PTC) that causes premature protein truncation and triggers a nonsense-mediated decay (NMD) process. Several recent studies have detected copy number variants (CNVs) in patients with FVER [15,16,17]. Taken together, these findings indicate that most *KIF11* variants cause *KIF11*-associated retinopathy through a loss of function mechanism. At present, however, no discernible genotype–phenotype correlation has been established in *KIF11*-associated retinopathy, as some variant carriers show mild effects or no clinical features, suggesting incomplete penetrance and variable expressivity for the phenotype [12,13,14].

In the current study, we have described the genetic and clinical characteristics of 10 individuals from 7 unrelated families with *KIF11* variants. We discovered 6 novel variants, comprising one missing allele, by whole genome sequencing (WGS). Minigene analysis verified one deep intronic (DIV) variant that caused the insertion of a pseudoexon (PE) and one synonymous variant that led to an exon skip. We have also described the clinical characteristics of the patients.

## 2. Materials and Methods

### 2.1. Patients

This study was approved by the Beijing Tongren Hospital Joint Committee on Clinical Investigation and was conducted in accordance with the tenets of the Declaration of Helsinki. Informed written consent was acquired from all patients or their guardians before enrollment in the investigation. In total, 10 individuals from 7 unrelated pedigrees who carried a heterozygous *KIF11* variant were enrolled in the Genetics Laboratory of the Beijing Institute of Ophthalmology. Of the seven probands, six were from 169 probands diagnosed with FVER. The diagnostic criteria for FVER included avascular periphery or anomalous intraretinal vascularization, retinal folds, subretinal exudation, vitreous hemorrhage, and RD [8]. In addition, all the patients were full-term births. The one remaining proband was from a group of 2003 probands clinically diagnosed with inherited retinal degeneration (IRD) who had undergone molecular genetic analysis. After collecting the medical and family histories, the syndromic disease manifestations were recorded. Microcephaly was defined as an occipitofrontal circumference (OFC) two or more standard deviations (SD) below the mean (≤−2 SD) for age and sex, based on World Health Organization (WHO) growth standards data. All individuals, if they could cooperate, underwent routine ophthalmic assessments that included best-corrected visual acuity (BCVA), slit-lamp biomicroscopy, and dilated fundus examination. Some cases underwent retinal optical coherence tomography (OCT) examinations. The six probands diagnosed with FEVR and their parents or related patients underwent wide-field fluorescence angiography and color fundus photography with RetCam III imaging under general anesthesia or using an Optos 200T× instrument (Optos, Dunfermline, UK).

### 2.2. Targeted Exon Sequencing (TES) and Whole Genome Sequencing (WGS)

Peripheral venous blood samples were collected from each participant and their available family members. Genomic DNA was extracted from peripheral leukocytes using a Whole Blood Genomic DNA Extraction Kit (Vigorous, Beijing, China) according to the manufacturer’s protocol. The seven probands underwent TES using a panel containing 533 known IDR genes. The capture panel design, Illumina library preparation, and capture experiment were performed as previously described [18]. We performed WGS using MGI, as previously described [19], to detect the missing variants in proband 0151230, who had both FEVR and microcephaly but whose TES did not identify any *KIF11* variant. Briefly, the WGS libraries were sequenced on the DNBSEQ-T7 platform (MGI Tech) using a 100 bp paired-end mode with a 30-fold minimal median coverage per genome [19].

### 2.3. Bioinformatics Analysis

Multiple in silico tools were used for variant annotation based on the type of variant identified. Missense variants were evaluated using three online bioinformatics analysis programs: Mutation Taster, PolyPhen-2, and SIFT. Five algorithms, namely Human Splicing Finder (version 2.4.1), Alternative Splice Site Predictor (version 2011-10-01), MaxEntScan (version 2003-7-22), NetGene2 (version 2.42), and NNSplice (version 0.9), were used to evaluate the probability that noncoding region variants or synonymous variants would generate abnormal transcript splicing. Variants were chosen for additional analysis by in vitro minigene assays when they were situated at putative splice sites with a comparative strength of at least 60% of the maximal score in at least two out of five splice prediction programs or when they led to an increased splice prediction score of at least 30% compared to the wild-type. All *KIF11* variants (NM_004523) considered to be disease-causing were validated by Sanger sequencing. Co-segregation analyses were performed in all the pedigrees.

### 2.4. Ultra-Deep Sequencing

Ultra-Deep sequencing was performed in the DNA samples extracted from peripheral venous blood to detect possible mosaicism in five pedigrees whose co-segregation analysis revealed no parents carrying diseasing-causing variants in *KIF11*. Amplicons covering five pathogenic variants (c.387G>A, c.934C>T, c.1875+42A>G, c.2068C>T, and c.2765del) were generated with the primers itemized in Appendix A. The library was loaded onto the Illumina HiSeq2000 for the initiation of cluster generation and a pair-end 150-cycle sequencing protocol. The median coverage was greater than 10,000× for all samples (15,958× to 318,693× depending on the amplicon; average 113,364× ± 117,948× SD). The proportion of mosaicism was calculated based on quantitative alternate allele frequency (AAF) [20]. If an AAF was below 0.5%, it was defined as the background.

### 2.5. Minigene Assay in HEK293T Cells

A novel DIV c.1875+42A>G and one novel synonymous variant c.387G>A were predicted to cause splicing abnormalities by a panel of algorithms; therefore, the two variants were assessed for their influence on splicing with the pET01-based exon trapping system (Exontrap, MoBiTec GmbH, Goettingen, Germany). The two cloned fragments, embracing the DIV c.1875+42A>G (exon 14) and c.387G>A (exon 4) as well as 200–1000 nucleotides of flanking exon or introns, were PCR amplified from the genomic DNA of patients 0151230 and 0191139, respectively, with the primers itemized in Appendix A.

The authentication of HEK293T cells was confirmed by short tandem repeat assays (Appendix A). The cells were transfected with 2.5 µg of the selected minigene plasmids (MP) using Lipofectamine 2000 DNA transfection reagent (Invitrogen, Carlsbad, CA, USA). After 48 h, the transfected cells were harvested, and total RNA was extracted using an FastPure Total RNA Isolation Kit (Vazyme, Nanjing, China). First, cDNA strand was synthesized using the FastKing gDNA Dispelling RT SuperMix (Tiangen Biotech, Beijing, China). For the analysis of splicing events, first-strand cDNA templates were used to amplify the region of interest with specific primers (ETPR04: 5′-GGATTCTTCTACACACCC-3′, ETPR05: 5′-TCCACCCAGCTCCAGTTG-3′). The amplification reaction underwent 35 cycles of 94 °C for 30 s, 60 °C for 30 s, and 72 °C for 30 s, followed by 72 °C for 5 min. The products were separated by electrophoresis on 2% agarose gels, excised, and sequenced.

## 3. Results

### 3.1. KIF11 Variants Detected

We identified 6 novel distinct causative *KIF11* variants in this cohort (Table 1). TES initially detected 5 variants, consisting of two nonsense, one frameshift, one missense, and one synonymous variant. Further WGS uncovered a missing DIV in one unsolved family 0151230. None of the variants were documented in any public database, and all were classified as pathogenic or likely pathogenic according to the American College of Medical Genetics and Genomics guidelines and standards (Table 1). Co-segregation analysis revealed that five variants (c.387G>A, c.934C>T, c.1875+42A>G, c.2068C>T, and c.2765del) were not identified in the parents of five probands. Subsequent ultra-deep sequencing found that the five variants were either absent or were present at very low AAFs (0.002) in the five unaffected parents, whereas the AAFs in the five probands were between 0.465 and 0.512 (Table 2).

### 3.2. A Novel DIV and a Novel Synonymous Variant Validated by Minigene Assays

The novel heterozygous DIV c.1875+42A>G and the synonymous variant c.387G>A met the inclusion criteria of the minigene assay. Bioinformatic analyses predicted that the synonymous variant c.387G>A, located at the last base of exon 4, altered the wild-type donor site, resulting in the skipping of exon 4, whereas the DIV triggered a cryptic splice donor site and caused the insertion of one PE (Appendix A). The abnormal splicing products are presumed to generate a PTC, causing the loss of the reading pattern and potentially resulting in mRNA degradation caused by NMD.

RT-PCR analysis revealed that the mutant c.387G>A construct generated an aberrant transcript (P2) corresponding to the exon 4 skip (Figure 1B). The synonymous variant erased the natural donor site and was predicted to form a stop codon at the sixth amino acid downstream in p.[Tyr104Ilefs*6]. In addition, the normal splicing band (P1) was faintly identified in the mutant construct (Figure 1B). The RT-PCR results indicated that the mutant c.1875+42A>G generated an abnormal splicing band (P3). The P3 contained a 42 nucleotide PE, which was predicted to encode a truncated protein p.[Asn626Valfs*14] (Figure 1C).

### 3.3. Clinical Findings

In the current cohort, 10 individuals (7 females and 3 males) from 7 unrelated pedigrees carried one heterozygous pathogenic KIF11 variant (Figure 2). The 7 probands included 6 sporadic cases and one (0151680) had an affected sister with FEVR. Co-segregation analysis detected KIF11 variants in two asymptomatic mothers (0151322 and 0151682) and one affected sister (0151683), but the remaining five probands carried a de novo variant (Table 3). All the probands were clinically diagnosed with FEVR, except for proband 0191139, who was initially diagnosed with IRD. Therefore, the prevalence of KIF11 variants in our FVER patients was 3.6% (6/169). The median age of the individuals with KIF11 variants at the last examination was 4 (range, 1–32) years.

In the current cohort, congenital microcephaly, ranging from 2 SD to 3 SD below the mean, was observed in all seven probands. Asymptomatic mother 0151322 had an OFC at the lower limit of normal. All the individuals exhibited special facial features, which included upslanting palpebral fissures, broad nose with rounded tip, and long philtrum with thin upper lip. Mild to moderate intellectual disability or developmental delay was detected in five probands, as was mild primary lymphedema at the lower legs and feet, but this later resolved and was noted in only proband 015460 (Table 3).

### 3.4. Ocular Phenotype

In the present cohort, eight patients (7 probands and one affected sister) showed different extents of visual defects. All the probands had nystagmus and suffered severe visual defects (hand moving to 0.1), while the affected sister 0151683 scored 0.8 in the right eye and 0.05 in the left eye. The two asymptomatic mothers displayed an almost normal BCVA (0.8–1.0). Four probands presented with cataracts at very early ages (1–3 years); one also had esotropia and one also had microcornea (Table 3). Fundus examinations revealed that all 10 individuals (20 eyes) with KIF11 variants exhibited different extents of abnormal fundus appearance (Figure 3, Figure 4, Figure 5 and Figure 6). Four patients showed an asymmetric fundus appearance between both eyes. Proband 0191139, who was initially diagnosed with Leber congenital amaurosis, presented with a large well-demarcated chorioretinal atrophy in the macular region in the right eye and optic disc dysplasia and retinal fold in the left eye (Figure 5D). In this cohort, retinal folds were observed in 10 eyes, followed by tractional RD detected in 9 eyes, chorioretinal dysplasia or atrophy in 8 eyes, and increased or straightening of the peripheral vessels or peripheral avascular zones in 7 eyes (Table 3). Asymptomatic mother 0151322 displayed bilateral tortuous vessels and posterior hyaloidal organization (Figure 6B).

## 4. Discussion

In this study, we performed a full genetic analyses and reported the clinical findings in a Chinese cohort with *KIF11*-related retinopathy from a tertiary center. We uncovered the missing heritability in one family by further WGS analysis. The frequency of the *KIF11* variant in FEVR was 3.6% (6/169), which was similar to the rate reported from a large Chinese cohort with FEVR that included 696 probands [21].

In line with previous observations [3,6,15,17], the six novel *KIF11* variants included only one missense variant; the remaining five were either nonsense or frameshift small indels or variants affecting splicing. DIV c.1875+42A>G, described here, is the first DIV identified in the *KIF11* gene. Our in vitro functional analysis showed that this DIV c.1875+42A>G stimulated a cryptic splice donor site to yield an abnormal splicing product that included the insertion of a PE. By contrast, the novel synonymous variant c.387G>A, located in the last base of exon 4, caused skipping of the entire exon 4 by erasing a wild-type donor site. This finding was similar to the observation in a synonymous variant located in exon 20 [22]. Both the PE insertion and exon 4 skipping resulted in alterations in the *KIF11* reading frame and finally led to a PTC. Our results further verified that the loss of KIF11 function was a mechanism underlying *KIF11*-related retinopathy. Our RT-PCR results also detected a very weak normal product band in the synonymous variant (c.387G>A) constructer; this was similar to our previous results for DIVs in two other IRD genes, *ABCA4* and *WFS1* [19,23].

Our initial co-segregation using Sanger sequencing and subsequent ultra-deep sequencing demonstrated that none of the five heterozygous variants of *KIF11* were detected in five unaffected parents, suggesting that these variants identified in the five sporadic pedigrees arose de novo. Therefore, the frequency of patients carrying a de novo *KIF11* variant was 71% (5/7), which was much higher than the 31% reported from a large cohort that included 52 families [12]. Our high frequency was due to the small family number and because six of the seven families were sporadic. In addition, compared to Sanger sequencing, ultra-deep sequencing might be more capable of detecting a mosaic variant that arises at a low rate. Karjosukarso et al. identified a mosaic variant at a frequency of 16.9% in blood using ultra-deep sequencing [15].

The patients in the current cohort presented with high variations in their phenotypes, and two families exhibited incomplete penetrance. All the probands showed more profound clinical features than their other affected family members. The characteristic facial features initially described in patients with MCLMR were observed in all patients, including the two asymptomatic mothers, whereas microcephaly was only noted in 70% of the patients; this rate was lower than the 90% reported in a previous study that included 87 patients [12]. Consistent with the previous observation, half of the patients in the current cohort had an intellectual disability or developmental delay; however, only one proband experienced primary lymphedema in her lower limbs, and this rate (10%) was much lower than the 47% noted in that large cohort that included 87 patients. The lymphedema in MCLMR usually spontaneously resolves in early childhood; therefore, it could be ignored by some parents [3].

Our comprehensive ophthalmic evaluations revealed that all the *KIF11* variant carriers had retinal anomalies. Retinal folds and RD were observed in all the patients, except for two asymptomatic mothers. Patients with RD or retinal folds usually suffered from severe visual impairments. Chorioretinal dysplasia or atrophy was found in 40% (8/20) of the eyes, but demarcated chorioretinal atrophy was only observed in two eyes. In a recent study, Wang et al. reported detecting chorioretinal dysplasia in 44.2% (31/70) of the eyes with *KIF11*-associated retinopathy, whereas this rate was only 1.3% (1/70) for patients with FEVR who carried variants in other genes. They stated that chorioretinal dysplasia was the principal retinal feature in *KIF11*-associated retinopathy [21]. The two asymptomatic mothers displayed either an increase and straightening of peripheral vessels or tortuous vessels and posterior hyaloidal organization. These were all mild retinal changes of FVER that did not affect their visual acuity [24]. In the current cohort, four patients from two unrelated families harbored the synonymous variant c.387G>A; however, they exhibited both obvious interfamily and intra-family phenotype variation. This suggested that phenotype expression might be related to other circumstances, such as environmental and epigenetic factors.

Our study had some limitations. One was its retrospective nature, and another was the small number of patients enrolled in our study. In addition, we did not perform quantitative analysis of the RT-PCR products of the synonymous variant; therefore, we could not decide on or calculate the mutant pathogenic strengths caused by this variant.

In conclusion, our results expand the pathogenic variant spectrum of *KIF11*. WGS disclosed the missing heritability of a proband whose variant was not located in the exons of *KIF11*. The DIVs elucidated the rare unsolved Chinese cases with *KIF11*-related retinopathy. Patients carrying *KIF11* variants might exhibit variable phenotypes and different expressions; therefore, genetic analysis is critical for the precise diagnosis of young or sporadic patients.

## Figures and Tables

**Figure 1 genes-14-00212-f001:**
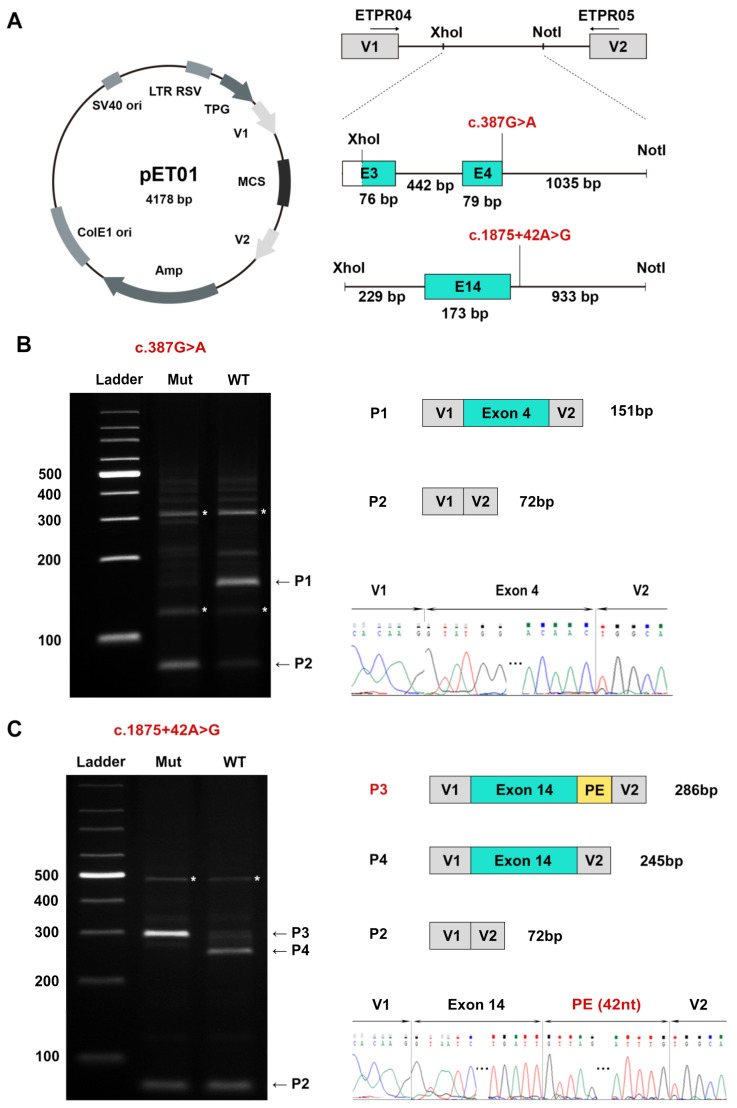
Results of minigene assays for variants c.387G>A and c.1875+42A>G identified in the *KIF11* gene. The target fragments were cloned into the multiple cloning site (MCS) of pET01 between exons V1 and V2. After transfection of HEK293T cells with the wild-type (WT) and mutant-type (Mut) constructs, RNA was isolated and subjected to RT-PCR. The generated cDNA was amplified using primers ETPR04 and ETPR05 within the exons V1 and V2. (**A**) Schematic representation of the location of splice variant sites and the fragments inserted into the pET01 construct. Note that exon 3 was not included in full. (**B**) Agarose gel electrophoresis of the amplified cDNA obtained from HEK293T cells transfected with c.387G>A mutant construct (Mut) and wild-type construct (WT). The WT band (P1) is consistent with higher molecular weight compared to the truncated form (P2) induced by exon skipping. (**C**) Agarose gel electrophoresis of cDNA amplification obtained from HEK293T cells transfected with c.1875+42A>G mutant construct (Mut) and wild-type construct (WT). The inclusion of a pseudoexon was observed in the Mut band (P3), but not in the WT transcript (P4). Subsequent Sanger sequencing confirmed the insertion of 42 bp from intron 14 due to the cryptic donor site activation. Gray boxes represent pET01 resident exons V1 and V2, and green boxes represent the exons from the *KIF11* gene. The yellow box represents the retained intronic sequence. Bands with asterisks indicate that no sequence data were obtained.

**Figure 2 genes-14-00212-f002:**
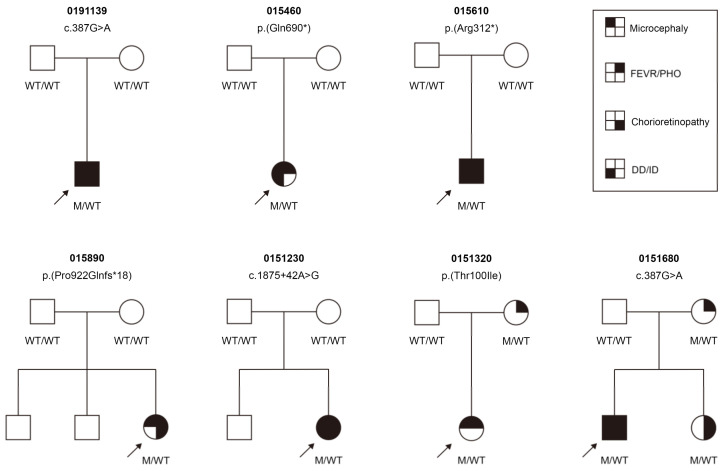
Pedigrees and phenotypes of families with mutations in the *KIF11* gene. DD, Development delay; FEVR, familial exudative vitreoretinopathy; ID, Intellectual disability; M, mutant-type; PHO, posterior hyaloidal organization; WT, wild-type.

**Figure 3 genes-14-00212-f003:**
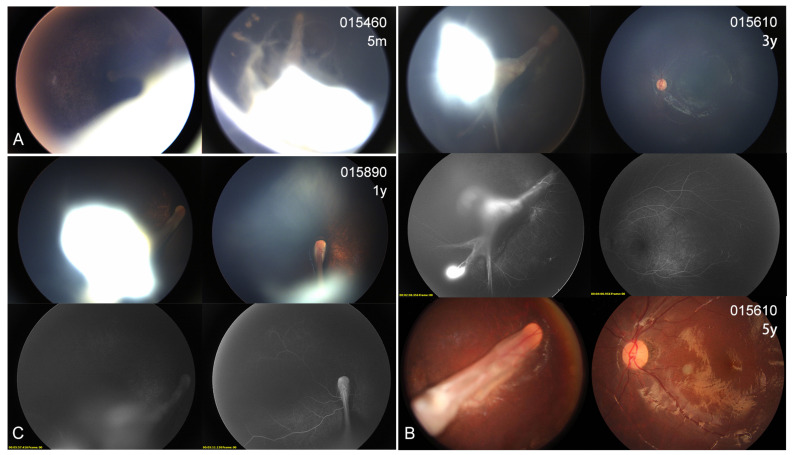
Color fundus (CF) and fluorescein angiography (FA) images of probands with KIF11 variants. (**A**) CF photographs of patient 015460 show bilateral tractional retinal detachment (RD) in both eyes. (**B**) CF and FA photographs of patient 015610 show a retinal fold (RF) involving the macula, tractional RD, and chorioretinal dysplasia in the right eye and peripheral avascular zones in the left eye. At the 2-year follow-up, the lesions appeared stable. (**C**) CF and FA images of patient 015890 show bilateral RFs and tractional RD in both eyes, in keeping with diffuse chorioretinal dysplasia.

**Figure 4 genes-14-00212-f004:**
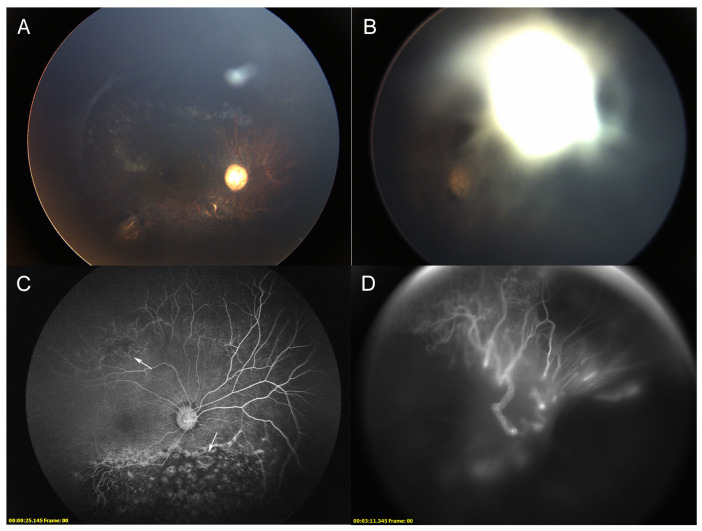
Color fundus (CF) and fluorescein angiography (FA) photographs of patient 0151230. CF photographs show patches of chorioretinopathy predominately located on the inferior retina and marked vascular attenuation in the right eye (**A**), and a tractional RD in the left eye (**B**). FA images highlight poorly developed retinal vasculature (**C**,**D**).

**Figure 5 genes-14-00212-f005:**
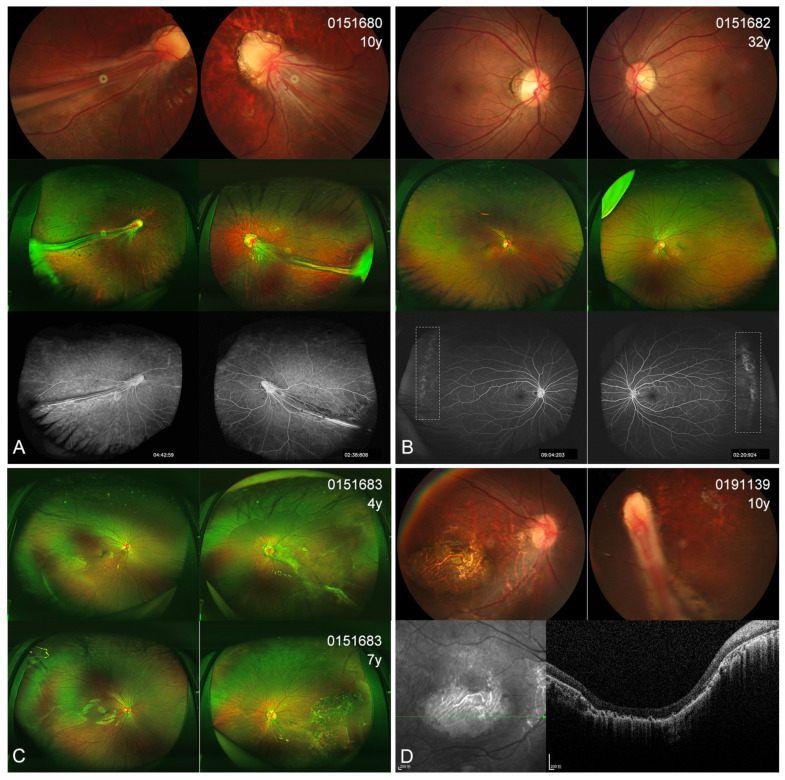
Color fundus (CF), scanning laser ophthalmoscopy (SLO), fluorescein angiography (FA) images, and OCT scans of four patients carrying the variant c.387G>A. (**A**) The CF, SLO, and FA photographs of proband 0151680 show bilateral retinal folds, increases in or straightening of the peripheral vessels (ISPV), and chorioretinal dysplasia. (**B**) The CF and SLO images of mother 0151682 display increased vascular branching, and the FA photographs indicate peripheral avascular zones (white dashed box). (**C**) SLO photographs of sister 0151683 show ISPVs in the right eye, as well as epiretinal membrane and tractional retinal detachment (RD), along with chorioretinal dysplasia in the left eye. Follow-up images 3 years later show that the laser demarcation of an inferotemporal RD on the left eye remained stable. (**D**) CF photographs of patient 0191139 show macular atrophy and a temporally dragged disc in the right eye and a retinal fold in the left. The OCT scan demonstrates severe focal loss of the chorioretina.

**Figure 6 genes-14-00212-f006:**
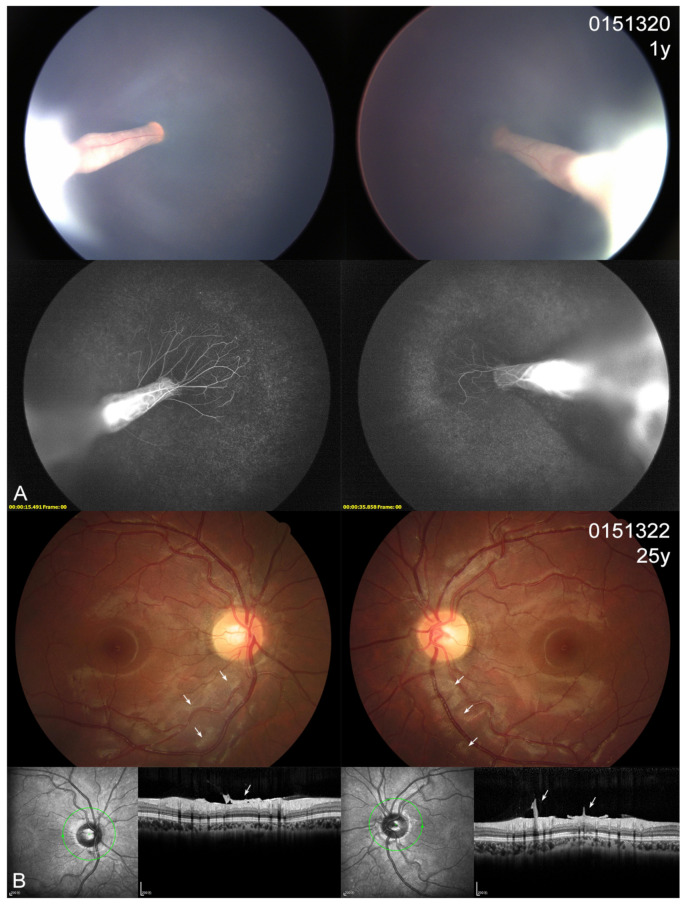
Color fundus (CF), fluorescein angiography (FA) photographs and OCT scans from a family with the variant c.299C>T. (**A**) The CF and FA images of proband 0151230 show bilateral retinal folds and tractional retinal detachments. (**B**) CF photographs and OCT scans of mother 0151322 display tortuous vessels and a thicker posterior hyaloidal matrix (white arrows).

**Table 1 genes-14-00212-t001:** Novel *KIF11* variants identified in this study and analysis of the variants by predictive.

Exon/ Intron	Nucleotide Change	Predicted Protein Effect ^#^	Allele Number	Type	MT	PP2	SIFT	HSF	MES	SpliceAI	MAF *	ACMG	Source
3	c.299C>T	p.(Thr100Ile)	1	Missense	DC	D	D	-	-	-	-	LP	Novel
4	c.387G>A	p.(=)	2	Splicing	DC	-	-	SC	SC	SC	-	P	Novel
8	c.934C>T	p.(Arg312*)	1	Nonsense	DC	-	-	-	-	-	-	P	Novel
IVS14	c.1875+42A>G	p.(Asn626Valfs*14)	1	Splicing	DC	-	-	SC	SC	SC	-	P	Novel
16	c.2068C>T	p.(Gln690*)	1	Nonsense	DC	-	-	-	-	-	-	P	Novel
19	c.2765del	p.(Pro922Glnfs*18)	1	Frameshift	DC	-	-	-	-	-	-	P	Novel

^#^ Coding sequence change was based on DNA reference sequence NM_004523 (*KIF11*). * Querying minor allele frequency (MAF) by three databases: the Exome Aggregation Consortium (ExAC), the Genome Aggregation Database (gnomAD), and the 1000 Genomes Project. Abbreviations: ACMG, classification of variants according to American College of Medical Genetics; DC, Disease causing; D, Damaging; HSF, human splicing finder; LP, likely pathogenic; MAF, minor allele frequency; MES, MaxEntScan; MT, Mutation Taster; P, pathogenic; PP2, Polyphen-2; SC, Splice site changed; SIFT, Sorting Intolerant From Tolerant.

**Table 2 genes-14-00212-t002:** The alternate allele frequency and average sequencing depth in all samples.

Genome Position *	Ref	Alt	Family ID	Family Member	Ref/Alt Reads	AAF	Depth ^#^
chr10:94397210	C	T	015460	proband	98,965/96,841	0.495	226458.81
			015461	father	184,385/0	0.000	213769.63
			015462	mother	152,119/0	0.000	176279.86
chr10:94373278	C	T	015610	proband	202,644/203,822	0.501	318693.21
			015611	father	350,057/0	0.000	274592.15
			015612	mother	354,036/0	0.000	277840.26
chr10:94408183-94408184	TC	T	015890	proband	10,244/8895	0.465	16783.04
			015891	father	18,053/0	0.000	15957.56
			015892	mother	21,111/0	0.000	17986.41
chr10:94393378	A	G	0151230	proband	15,908/15,686	0.494	26361.03
			0151231	father	15,717/33	0.002	21791.67
			0151232	mother	32,161/43	0.001	27411.65
chr10:94366994	G	A	0191139	proband	17,506/18,402	0.512	23877.27
			0191139-1	father	43,078/79	0.002	30039.17
			0191139-2	mother	48,655/99	0.002	32621.80

* Sequence data were aligned against the human reference genome (hg19). ^#^ Depth means average sequencing depth on target region. Abbreviations: AAF, alternate allele frequency; Alt, Alternative Allele; Ref, Reference Allele.

**Table 3 genes-14-00212-t003:** Clinical data of individuals carrying pathogenic variants in *KIF11*.

Patient ID	Gender	Exam Age (Year)	BCVA (OD/OS)	Nyst- Agmus	Ocular Manifestations (OD/OS or OU)	Micro- Cephaly	Lymph- Edema	DD/ID	CFP	Variants	Co- Segregation
0191139	M	10	NA	+	dragged-disc, chorioretinal atrophy/esotropia, RF	+	-	+	+	c.387G>A, p.(=)	De novo
015460	F	<1	NA	+	cataract, tractional RD	+	+	+	+	c.2068C>T, p.(Gln690*)	De novo
015610	M	3	HM/0.1	+	cataract, RF, tractional RD, chorioretinal dysplasia/peripheral avascular zones	+	-	+	+	c.934C>T, p.(Arg312*)	De novo
015890	F	3	NA	+	RFs and tractional RD, chorioretinal dysplasia	+	-	-	+	c.2765del, p.(Pro922Glnfs*18)	De novo
0151230	F	2	NA	+	chorioretinal dysplasia, peripheral avascular zones/microcornea, cataract, tractional RD	+	-	+	+	c.1875+42A>G, p.(Asn626Valfs*14)	De novo
0151320	F	1	NA	+	cataract, RFs, tractional RD	+	-	-	+	c.299C>T, p.(Thr100Ile)	Maternal
0151322	F	25	1.0/1.0	-	tortuous vessels, posterior hyaloidal organization	-	-	-	+	c.299C>T, p.(Thr100Ile)	
0151680	M	10	0.05/0.05	+	RFs, increase or straightening of peripheral vessels, chorioretinal dysplasia	+	-	+	+	c.387G>A, p.(=)	Maternal
0151682	F	32	1.0/0.8	-	increase or straightening of peripheral vessels	-	-	-	+	c.387G>A, p.(=)	
0151683	F	4	0.8/0.05	-	increase or straightening of peripheral vessels/ERM, tractional RD, chorioretinal dysplasia	-	-	-	+	c.387G>A, p.(=)	

Abbreviations: BCVA, best corrected visual acuity; CFP, characteristic facial phenotype; DD, development delay; ERM, Epiretinal membrane; FFA, fundus fluorescein angiography; HM, hand motion; ID, intellectual disability; LP, light pursuit; NA, not available; NLP, no light perception; OD, right eye; OFC, occipitofrontal circumference; OS, left eye; OU, bilateral eyes; RD, retinal detachment; RF, retinal fold.

## Data Availability

All raw data used during the study are available from the corresponding author by request.

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
