# Peer review of "Phenotype-Based Genetic Analysis Reveals Missing Heritability of KIF11-Related Retinopathy: Clinical and Genetic Findings"

_genes, 2023, doi:10.3390/genes14010212_

Round 1

Reviewer 1 Report

1- Citations throughout the text should be revised, as the authors added the references after the period and should be at the end of the sentence but before the period. For example, in line 36, the authors wrote: vascular system function.[1,2] and it should be vascular system function [1,2].                                                          Please revise the citations with the correct format in the entire manuscript. 

2- In the experiment Minigene assay, why the  pair of primers used for the RT-PCR is not given in the supplemental tables? also why the PCR condition is not written?

3- In the gel image Figure 1B : what are the upper faint bands in the mutant and WT above the 300 bp? can you elaborate on what these bands might be? 

4- The author mention in line 199, figure 1 legend: "Bands with asterisks indicate that no sequence data were obtained" Why the sequence failed for these bands? I think you should have tried to optimized  the sequencing protocol to have them sequenced successfully. 

5- scale bars for the images in figures 3,4, 5 and 6 are missing

Author Response

We would like to thank you for the constructive comments and careful corrections regarding our manuscript “Phenotype-Based Genetic Analysis Reveals Missing Heritability of KIF11-Related Retinopathy: clinical and genetic findings” (ID: genes-2116459). Based on your suggestions, we have addressed each of your comments as indicated in our point-by-point response below. All changes are in blue.

Point 1: Citations throughout the text should be revised, as the authors added the references after the period and should be at the end of the sentence but before the period. For example, in line 36, the authors wrote: vascular system function.[1,2] and it should be vascular system function [1,2]. Please revise the citations with the correct format in the entire manuscript.  

Response 1: Thank you for your kind reminder. We have modified the citations throughout the manuscript with the correct format, please check the new version of revised manuscript.

Point 2: In the experiment Minigene assay, why the pair of primers used for the RT-PCR is not given in the supplemental tables? also why the PCR condition is not written?

Response 2: We added primer sequences and reaction conditions used in the RT-PCR process based on your advice, please refer Line 150-154.

Point 3: In the gel image Figure 1B : what are the upper faint bands in the mutant and WT above the 300 bp? can you elaborate on what these bands might be?

Response 3: We got your concern. The minigene assay might had different splicing products and we speculated these bands were alternative splicing products, however, the bands were too faint to get an accurate sequence.

Point 4: The author mention in line 199, figure 1 legend: "Bands with asterisks indicate that no sequence data were obtained" Why the sequence failed for these bands? I think you should have tried to optimized the sequencing protocol to have them sequenced successfully.

Response 4: We have managed to sequence the bands with several different conditions, unfortunately, we did not get any clean sequences related with KIF11, therefore, we denotes them as the fragment for which no sequence information was obtained. We will optimized the sequencing protocol to have them sequenced successfully in the future.

Point 5: scale bars for the images in figures 3,4, 5 and 6 are missing.

Response 5: It is usually no need to have a scale bar in fundus photography, this is different from HE stained or immunofluorescence images. Diameter of optic disc is about 1.5mm, we can roughly estimate size of lesion. Please refer papers (doi:10.3390/genes13040713; doi: 10.3390/ijms22105396).

Once again, we would like to thank you for the helpful comments and hope our responses adequately address your concerns.

Reviewer 2 Report

The manuscript is an additive knowledge for genetic analysis of KIF11 gene, which is critical for diagnosis and screening. The manuscript is well written and accounted for new DIV incorporation into the study. 

The manuscript can be accepted in the current format

Author Response

We feel great thanks for your professional review work and kind comments on our article.

Reviewer 3 Report

Congratulations to the authors. Well-designed, consistent, well-written work and a very relevant topic to understand unresolved cases of FEVR.

Author Response

We sincerely thank you for your nice comments on our article.